# Idiosyncratic Drug-Induced Liver Injury: Mechanistic and Clinical Challenges

**DOI:** 10.3390/ijms22062954

**Published:** 2021-03-14

**Authors:** Alison Jee, Samantha Christine Sernoskie, Jack Uetrecht

**Affiliations:** 1Department of Pharmacology and Toxicology, University of Toronto, Toronto, ON M5S 1A8, Canada; alison.jee@mail.utoronto.ca; 2Department of Pharmaceutical Sciences, University of Toronto, Toronto, ON M5S 3M2, Canada; sam.sernoskie@mail.utoronto.ca

**Keywords:** liver injury, adverse drug reactions, immunotoxicity, innate immune response, damage-associated molecular pattern molecules, reactive metabolites, cytochromes P450

## Abstract

Idiosyncratic drug-induced liver injury (IDILI) remains a significant problem for patients and drug development. The idiosyncratic nature of IDILI makes mechanistic studies difficult, and little is known of its pathogenesis for certain. Circumstantial evidence suggests that most, but not all, IDILI is caused by reactive metabolites of drugs that are bioactivated by cytochromes P450 and other enzymes in the liver. Additionally, there is overwhelming evidence that most IDILI is mediated by the adaptive immune system; one example being the association of IDILI caused by specific drugs with specific human leukocyte antigen (HLA) haplotypes, and this may in part explain the idiosyncratic nature of these reactions. The T cell receptor repertoire likely also contributes to the idiosyncratic nature. Although most of the liver injury is likely mediated by the adaptive immune system, specifically cytotoxic CD8+ T cells, adaptive immune activation first requires an innate immune response to activate antigen presenting cells and produce cytokines required for T cell proliferation. This innate response is likely caused by either a reactive metabolite or some form of cell stress that is clinically silent but not idiosyncratic. If this is true it would make it possible to study the early steps in the immune response that in some patients can lead to IDILI. Other hypotheses have been proposed, such as mitochondrial injury, inhibition of the bile salt export pump, unfolded protein response, and oxidative stress although, in most cases, it is likely that they are also involved in the initiation of an immune response rather than representing a completely separate mechanism. Using the clinical manifestations of liver injury from a number of examples of IDILI-associated drugs, this review aims to summarize and illustrate these mechanistic hypotheses.

## 1. Introduction

Idiosyncratic drug-induced liver injury (IDILI) represents a major problem for multiple stakeholders. It has surpassed viral hepatitis as a major cause of acute liver failure in North America [1]. It also adds a significant risk to drug development because it is only detected late in phase III clinical trials or even after a drug has been marketed. It is likely that a, if not the, major cost of drug development is the need to treat thousands of patients in clinical trials to ensure that a drug does not cause an unacceptable risk of idiosyncratic drug reactions, with IDILI being a major idiosyncratic reaction leading to drug candidate failure. Many attempts have been made to develop in vitro assays that predict the risk of a drug causing idiosyncratic reactions [2]. These assays also increase the cost and time required to bring drug candidates forward for clinical testing, but none have proven to be reliable. In vitro assays cannot duplicate the complexity of IDILI and, without a better understanding of the mechanisms of idiosyncratic drug reactions, it is unlikely that this situation will improve. Given its idiosyncratic nature, mechanistic studies of IDILI are very difficult. Animals can also develop IDILI, but it is also idiosyncratic in animals. Most of the animal models that have been studied represent acute toxicity caused by high doses of drug and do not represent the mechanism of IDILI. It is essential to start with the clinical characteristics of IDILI; any hypothesis that is inconsistent with these characteristics is very unlikely to be true. Thus, the goals of this review are to highlight current evidence for IDILI mechanistic hypotheses using a number of example drugs and to suggest how such hypotheses could be tested. These hypotheses may also allude to strategies that could be useful in the prevention or treatment of IDILI.

## 2. Clinical Characteristics

The clinical characteristics of IDILI provide important clues to the mechanisms underlying IDILI. The primary characteristics of IDILI are its idiosyncratic nature and the delay between starting the drug and the onset of evidence of liver injury. The typical time to onset is 1–3 months, but cases can occur slightly earlier, and some cases can occur after more than a year of therapy [3]. In some cases, the onset of IDILI can be more than 3 weeks after discontinuation of therapy [4]. There are exceptions: IDILI caused by telithromycin and fluoroquinolones can have a rapid onset on first exposure [5,6]. However, this is quite uncommon and may indicate that the mechanism in this case is significantly different. Classically, the time to onset is much shorter on rechallenge suggestive of immune memory, but that is not always the case, and it is not uncommon that rechallenge actually does not lead to liver injury [7]. These are the same characteristics as other idiosyncratic adverse drug reactions that affect other organs such as the skin and blood cells [8]. In many cases these other idiosyncratic adverse drug reactions are known to be immune mediated. There has been more reluctance to consider IDILI immune mediated, and IDILI was previously classed as either representing immune or metabolic idiosyncrasy [9]. IDILI is often classed as dose-independent; that is simply not true, and in general idiosyncratic drug reactions are uncommon with drugs given at a dose of 10 mg/day or less [10]. What is true is that the dose response curve for the therapeutic effects of a drug and the risk of IDILI may be in a different range, and there may not be a significant difference in risk within the therapeutic range of a drug.

The major risk factor that has been identified for IDILI caused by some drugs is the association with specific HLA haplotypes [11]. This is also true for other types of idiosyncratic drug reactions [12]. Additional genetic risk factors are polymorphisms in other immune-related genes such as a missense variant in protein tyrosine phosphatase non-receptor type 22 (PTPN22) [13] and a variant interleukin (IL)-10 allele [14]. It has been proposed that different environmental and disease states can affect the risk of IDILI. However, Hyman Zimmerman famously stated that pre-existing liver disease did not affect the risk of IDILI. One might expect that inflammatory conditions such as Crohn’s disease would increase the risk of IDILI, but if there is such an effect, it is not obvious. The microbiome does appear to affect the risk of immune checkpoint inhibitor-induced idiosyncratic reactions [15]. Undoubtedly, there are environmental and disease conditions that affect the risk of IDILI [16], but in most cases the evidence is not clear, and there do not appear to be obvious pre-existing conditions or environmental factors that predict IDILI risk.

IDILI can be classified as hepatocellular, cholestatic, or mixed. This classification is based on the ratio of alanine transaminase (ALT) to alkaline phosphatase expressed as times the upper limit of normal. If the ratio is greater than 5 it is considered hepatocellular; if less than 2 it is considered cholestatic; and if between 5 and 2 it is considered mixed [3,17]. Another type of IDILI is autoimmune DILI, which is similar to idiopathic autoimmune hepatitis with prominent autoantibodies and a predominance in females; however, the autoantibody profile is somewhat different [18]. The typical histology features interface hepatitis and a plasma cell infiltrate. The time to onset for autoimmune IDILI is usually longer than for other forms of IDILI, often more than a year, but it is different for different drugs [19]. The major type of IDILI leading to liver failure is hepatocellular IDILI; therefore, most of the discussion will be focused on this type of IDILI. Even though there are common features to IDILI caused by different drugs, there are also significant differences. These differences also provide important mechanistic clues, and this section illustrates some of these key nuances though the discussion of specific example drugs that are associated with the risk of IDILI.

### 2.1. Halothane

One of the first drugs to be associated with serious IDILI was halothane. It is used as a general anesthetic; therefore, unlike most drugs, exposure to halothane is relatively brief. More than 80% of the cases occur after more than one exposure, and in the few cases where it occurred after the first exposure, the time between exposure to halothane and the onset of liver injury was longer, i.e., from about 6 days to about 12 days [9]. There are also cases in which anesthesiologists developed halothane IDILI from occupational exposure to repeated, but much lower, levels of halothane. Even though liver injury does not usually occur with the first exposure, many patients do develop a fever on first exposure. In general, fever is mediated by cytokines, especially IL-6 and IL-1β, which is an indication of an immune response. Halothane is metabolized to a reactive metabolite: trifluoroacetyl chloride [20]. Isoflurane has a similar structure and is metabolized to the same reactive metabolite; however, the extent of reactive metabolite formation is less with isoflurane, and isoflurane is associated with a lower risk of IDILI than halothane [21]. This suggests that the trifluoracetyl chloride is responsible for halothane IDILI. Halothane-induced liver injury is often associated with eosinophilia, and sometimes with a rash [22]. These characteristics are usually taken as evidence for an immune mechanism. In addition, patients who developed halothane IDILI were found to have antibodies against trifluoroacetylated proteins as well as autoantibodies against native proteins including cytochromes P450 [23]. Children appear to be at a lower risk of halothane IDILI, and obesity appears to increase the risk [9]. It is likely that obese patients are exposed to a larger quantity of halothane, because halothane is quite lipophilic, and it requires a larger quantity of halothane to achieve the same level of halothane in the brain to induce anesthesia. The incidence also appears to be higher in women than men. Halothane is rarely used today because of safer alternatives.

The characteristics of halothane IDILI strongly suggest an immune mechanism in which the reactive metabolite produces drug-modified proteins, and in some cases, this results in an adaptive immune response leading to serious liver injury. The initial exposure is too short to develop a significant adaptive immune response, which requires expansion of the lymphocytes that are specific for the drug-modified proteins, but it is sufficient to produce memory T cells that can respond more rapidly on rechallenge. Attempts have been made to reproduce this liver injury in animals. A study by Furst et al., found that halothane produced significant liver injury in guinea pigs and a positive lymphocyte transformation test [24]. Repeated exposures appeared to lead to some degree of immune tolerance. More recently, a study by Chakraborty et al., found that depletion of myeloid-derived suppressor cells increased halothane-induced liver injury in mice and produced a reasonable animal model of halothane IDILI [25]. As the combination of clinical characteristics and animal data led to the conclusion that halothane IDILI is immune mediated, halothane IDILI was used as a classic example of “immune idiosyncrasy” [9].

### 2.2. Isoniazid

Another drug that is well known to cause IDILI is isoniazid. Due to the relatively high incidence of isoniazid IDILI, patients being treated with isoniazid are usually monitored with monthly ALT measurements to decrease the risk of liver failure. If a patient does develop an increase in ALT, if the ALT is not too high, it is usually possible to treat through it (a classic example of “adaptation”), and if the drug is stopped, it is usually possible to restart with a lower dose without causing serious liver injury [26]. However, rechallenge of a patient with a history of serious isoniazid IDILI can result in a very rapid onset of severe liver injury [27].

Studies in rats performed more than 4 decades ago found that high doses of isoniazid caused acute liver injury in rats, and this injury appeared to be due to bioactivation of N-acetylhydrazine, a metabolite of isoniazid [28]. However, given the acute nature of the liver injury in rats, it is likely that the mechanism in this animal model is different from isoniazid IDILI in humans. Due to the results in rats suggesting that N-acetylhydrazine was an essential intermediate in the bioactivation of isoniazid, studies were performed in humans to determine if acetylation phenotype was a risk factor for isoniazid IDILI. The first studies reported that patients with the rapid acetylator phenotype were at increased risk [29]. However, later studies suggested that, if anything, patients with the slow acetylator genotype are at increased risk of isoniazid IDILI, but the relationship remains controversial [30]. Furthermore, in mice and humans, the major reactive metabolite appears to involve direct oxidation of isoniazid to a reactive diazo metabolite [31].

As isoniazid IDILI is less commonly associated with fever, rash, or eosinophilia, and patients who developed isoniazid IDILI could usually be rechallenged without recurrence of liver injury, it was believed that isoniazid IDILI was not immune mediated [9]. The fact that it was initially believed that rapid acetylators were at increased risk led to isoniazid IDILI being characterized as “metabolic idiosyncrasy”. However, as mentioned, the association with acetylator phenotype is controversial, and if there is an effect it is too small to explain the idiosyncratic nature of isoniazid IDILI. In addition, the absence of fever, rash, and eosinophilia should not be used as evidence against an immune mechanism; most immune responses such as the response to viral hepatitis B are not associated with fever, rash, and eosinophilia. Subsequent studies have shown that patients who have a small increase in ALT also have an increase in Th17 cells and T cells expressing IL-10 [32]. In addition, most patients who develop isoniazid-induced liver failure have antibodies against isoniazid-modified proteins or native cytochromes P450 [33]. Furthermore, patients with isoniazid IDILI were found to have a positive lymphocyte transformation test [34] and isoniazid-specific T cells [35]. Although isoniazid IDILI is less frequently associated with eosinophilia than halothane IDILI, it is not rare [36], and isoniazid treatment is associated with other types of immune responses such as drug-induced fever and a drug-induced lupus-like autoimmune syndrome [37]. Taken together, these results strongly suggest that isoniazid IDILI is also immune mediated. Although isoniazid IDILI has been reported to be associated with the HLA-DQA1*0102 haplotype, the odds ratio was only 4, and the results have not been replicated [38]. The fact that there does not appear to be a strong HLA association may be because of the large number of proteins that are modified by the reactive metabolite of isoniazid as discussed later. This provides a large number of possible drug-modified peptides and increases the chance that there will be multiple HLA haplotypes that can recognize one of these peptides. In contrast, there is no example of IDILI in which a single genetic polymorphism in a metabolic pathway explains the idiosyncratic nature of IDILI caused by that drug. Interindividual differences in drug metabolism undoubtedly play a role in the idiosyncratic nature of IDILI, but it appears that it is a relatively small role [30]. On the other hand, if reactive metabolite formation required a specific enzyme, and a patient was deficient in that enzyme they should be protected. Such an occurrence is likely to be relatively rare, and most patients do not develop IDILI when treated with a drug; therefore, such a relationship would be hard to detect. It also would not explain the idiosyncratic nature of IDILI. Therefore, the term metabolic idiosyncrasy as a general category of IDILI should probably be abandoned.

### 2.3. Amodiaquine

Amodiaquine can cause agranulocytosis in addition to IDILI, and sometimes both in the same patient [39]. This is presumably because it is oxidized to an iminoquinone reactive metabolite by both cytochromes P450 in the liver [40], and also by myeloperoxidase in neutrophils and neutrophil precursors [41]. Anti-drug antibodies were observed in patients with amodiaquine idiosyncratic drug reactions including IDILI, which suggests that these idiosyncratic reactions are immune mediated [42]. However, the relatively high risk of idiosyncratic reactions associated with amodiaquine has limited it use, and to our knowledge, no studies have been performed to determine whether there is an association with specific HLA genotypes have been performed.

Most drugs that cause IDILI do not cause delayed-onset liver injury in animals at doses that result in what would be a therapeutic blood level in humans, but amodiaquine is an exception. Treatment of either mice or rats with amodiaquine leads to a delayed onset mild liver injury [43]. This injury appears to be mediated by NK or NKT cells because depletion of these cells is protective in mice. Impairment of immune tolerance by using a combination of PD-1^ȡ/−^ mice and anti-CTLA-4 markedly increases this delayed onset liver injury and produces an animal model with liver histology that closely mimics IDILI in humans [44]. PD-1 and CTLA-4 are immune checkpoint molecules that are expressed on T cells that limit immune responses to prevent damage to the host, but they can also prevent the immune system from destroying cancer cells [45]. Therefore, immune checkpoint inhibitors have been developed to treat cancer; however, they also increase the risk of IDILI as discussed later. Although the liver injury in this model does not result in liver failure, the injury is substantial, and there is an increase in bilirubin. The injury in this model involves CD8+ T cells, and depletion of these cells is protective [46]. We have used this model of impaired immune tolerance to explore other factors involved in IDILI [46,47,48,49,50,51]. In addition, the impaired immune tolerance model unmasks the ability of other drugs to cause liver injury; however, with the other drugs that have been tested, the liver injury is milder [52].

### 2.4. Nevirapine

Nevirapine can cause both IDILI and severe skin rashes. In this case, it appears that the ability of nevirapine to cause two different types of idiosyncratic reactions is because it can form two different reactive metabolites: a reactive quinone methide in the liver [53] and a benzylic sulfate in the skin [54]. The quinone methide is a suicide substrate for cytochromes P450, leading to initial inhibition followed by induction of P450 synthesis, while sulfotransferase in the skin forms 12-hydroxynevirapine sulfate, which can react with protein as the sulfate is a good leaving group [54]. Nevirapine also causes an immune mediated skin rash in female brown Norway rats, which is one of the few animal models of an idiosyncratic drug reaction that is similar to the idiosyncratic reaction that the drug causes in humans [55]. Nevirapine IDILI is presumably caused by the quinone methide. The HLA associations for nevirapine-induced serious skin rashes and liver injury are different. Specifically, several HLA haplotypes, including HLA-C*0401, are associated with an increased risk of toxic epidermal necrolysis [56], while HLA-DRB1*0101 is associated with an increased risk of nevirapine IDILI [57]. That is not surprising because the reactive metabolites are different, and the spectrum of proteins modified are likely to be different because they are formed in different tissues. Nevirapine is one of the drugs that can cause immune mediated liver injury in the impaired immune tolerance model [52].

### 2.5. Flucloxacillin

The dominant IDILI phenotype leading to liver failure is hepatocellular; however, some drugs are associated with a cholestatic IDILI phenotype. One such drug is flucloxacillin. Flucloxacillin IDILI is strongly associated with the HLA-B*5701 haplotype with an odds ratio of 80.6, which is compelling evidence that flucloxacillin IDILI is immune mediated [58]. In addition, activation of fluoxetine-specific T cells is HLA-B*5701 restricted [59]. However, even if a patient expresses that haplotype and is treated with flucloxacillin, there is less than a 1/500 chance that they will develop serious IDILI. One factor that probably influences what type of idiosyncratic drug reaction a drug causes is where the reactive metabolite is formed. In most cases, the reactive metabolite is formed by cytochromes P450 and binds to intracellular hepatic proteins. Therefore, the drug-modified proteins will be processed and presented on the cell surface in the context of MHC-I as discussed below. That is likely to lead to a CD8+ T cell response, which can result in hepatocellular damage. Therefore, why does flucloxacillin cause cholestatic liver damage? In contrast to most drugs, flucloxacillin is chemically reactive because it is a β-lactam, and it does not require bioactivation. In addition, flucloxacillin is a substrate for multidrug resistance-associated protein 2 (MRP2), which concentrates the drug in bile epithelia [60]. That is presumably why flucloxacillin causes a cholestatic form of IDILI. Other drugs that cause cholestatic IDILI such as terbinafine and chlorpromazine form reactive glutathione adducts that are likely to be substrates for transporters that concentrate them in bile epithelia [61,62].

### 2.6. Valproate

It has been proposed that one mechanism by which drugs can cause IDILI involves mitochondrial injury as discussed below. However, most IDILI is not associated with characteristics associated with mitochondrial injury (e.g., lactic acidosis). An exception is valproate IDILI, which is quite different from the IDILI caused by other drugs. There are essentially three somewhat different clinical pictures: the most frequent is chronically evolving liver failure with hepatic encephalopathy; less common is hyperammonemia with less evidence of hepatocellular injury; and least common is an acute Reye’s-like syndrome [9]. It is not clear whether these different clinical presentations indicate that the mechanisms involved are markedly different or whether there is a common mechanism that results in different clinical presentations in different patients. Hyperammonemia is common and out of proportion to the level of liver injury. Histology is most commonly characterized by microvesicular steatosis with varying degrees of necrosis.

The incidence of IDILI caused by many drugs, such as isoniazid and halothane, is higher in older patients, with children being relatively resistant. However, another unique feature of valproate IDILI is that the risk is higher in children, especially infants age 2 or younger [9]. Other risk factors are several inborn errors of metabolism such as medium chain acyl co-enzyme A (Co-A) dehydrogenase deficiency, and ornithine transcarbamylase deficiency, and polytherapy with anticonvulsants that induce cytochromes P450 [63]. Valproate IDILI often occurs after a febrile illness, which may put additional stress on metabolic pathways. Another risk factor for valproate IDILI that has been reported is a heterozygous genetic variation in mitochondrial DNA polymerase γ (POLG) [64]. This association has been used to exclude patients with this genotype from receiving valproate. However, this association has also been disputed [65]. Although the number of cases was too small to clearly show that this genetic variation is not a risk factor, the study by Hynynen et al., study does demonstrate that most patients with this genetic variation do not develop serious IDILI when treated with valproate.

The chemical structure of valproic acid can be characterized as dipropylacetic acid. The normal metabolism of fatty acids is β-oxidation, but the β-position is where valproate is branched, which would block oxidation to form acetate. Therefore, when valproate forms a Co-A ester it may block oxidation of fatty acids, and this is a plausible mechanism of the microvesicular steatosis [66]. It has also been postulated to deplete Co-A. Valproate is also oxidized to 4-ene and 2, 4-diene metabolites. The 4-ene metabolite is structurally very similar to hypoglycin A, which is the agent responsible for Jamaican vomiting sickness, a condition similar to Reye’s syndrome [9]. Reye’s syndrome also occurs almost exclusively in childhood and is associated with the use of aspirin for a febrile illness. Aspirin is another carboxylic acid that can damage mitochondria [67]. It can form a Co-A ester, but it cannot undergo β-oxidation analogous to a fatty acid. The 2,4-diene metabolite is chemically reactive and forms a glutathione adduct [68].

In summary, valproate IDILI is distinct from other IDILI. There are multiple lines of evidence that suggest it involves mitochondria. Valproate affects several metabolic pathways, and it very well may represent “metabolic idiosyncrasy” with no immune component. However, there does not appear to be a single genetic polymorphism that represents a clear cause of valproate idiosyncrasy, and it is still possible that valproate IDILI has an immune component.

### 2.7. Minocycline

Minocycline can produce typical hepatocellular IDILI, but it can also produce autoimmune IDILI [19]. The autoimmune-type of IDILI often occurs after a year or more of treatment and has a more chronic course. There is a marked female preponderance, and another risk factor is the HLA-B*3502 haplotype [69]. It is usually associated with antinuclear antibodies and elevated IgG serum levels [19]. The typical histology is characterized by the presence of plasma cells and interface hepatitis. Autoimmune IDILI usually responds well to corticosteroids and is less likely to recur than idiopathic autoimmune hepatitis when the steroids are stopped. Minocycline can also cause other types of autoimmune reactions such as a lupus-like syndrome [70]. Minocycline is structurally different from other tetracycline antibiotics in having a dimethylamine group para to a phenolic group, which permits it to be oxidized to a reactive metabolite [71]. Other drugs commonly associated with autoimmune IDILI are nitrofurantoin and α-methyldopa.

### 2.8. Biologics

Although biologics are generally expected to have fewer off-target effects due to their specific mechanisms of action, multiple biologics have been associated with IDILI. The small molecule drugs discussed above likely cause IDILI due to their bioactivation into reactive metabolites, but biologics may induce IDILI because biologics are themselves foreign proteins which may induce an immune response, or because the biologics exert effects on the immune system itself. Indeed, anti-drug antibodies have been identified during treatment with many biologics. However, these antibodies tend to be neutralizing, and the major concern is usually reduced efficacy, although hypersensitivity reactions may also be predicted by the detection of anti-drug antibodies [72]. Two classes of biologics are particularly implicated in IDILI, the immune checkpoint inhibitors and tumour necrosis factor-alpha (TNF-α) inhibitors, although other biologics also carry risk of causing hepatotoxicity, some of which are discussed below.

The immune checkpoint inhibitors have emerged as a cause of IDILI as their use has become widespread in oncology. Since checkpoint inhibitors disrupt immune tolerance, it is not surprising that immune-related adverse effects caused by checkpoint inhibitors can affect any organ; indeed, our own work discussed above has shown that liver injury is unmasked in animal models of checkpoint inhibition. The incidence of hepatotoxicity caused by checkpoint inhibitors varies depending upon the drug (or combination thereof), the dose, and the cancer being treated [73,74]. Anti-CTLA-4 IDILI appears more common than anti-PD-1 IDILI, while combination immunotherapy has the highest incidence [73]. One study found that increases in ALT were higher in patients being treated for hepatocellular carcinoma compared to those being treated for melanoma or non-small cell lung cancer, but there was no difference in discontinuation of therapy or death [75]. It is possible that this finding is due to the fact that those with hepatocellular carcinoma have underlying liver dysfunction; it could also be true that the increased ALT might even function as a marker of efficacy in this setting due to the destruction of malignant hepatocytes. Interestingly, while it seems reasonable that the disruption of immune tolerance by these therapies may lead to autoimmune reactions, hepatotoxicity caused by immune checkpoint inhibitors has both features that overlap with or are distinct from those of autoimmune hepatitis. For instance, liver injury associated with anti-PD-1 checkpoint treatment appears to lack certain features of autoimmune hepatitis, such as anti-nuclear antibodies, hypergammaglobulinemia, and a predominance in females [74]. Histologically, autoimmune hepatitis is often characterized by infiltrating plasma cells, while checkpoint inhibitor-induced liver injury is characterized by an infiltrate of predominantly histiocytes [74]. While autoimmune hepatitis and anti-PD-1 liver injury are characterized by both infiltrating CD4+ and CD8+ T cells, anti-CTLA-4 liver injury is predominantly characterized by CD8+ T cell infiltration [76,77]. It is also interesting to note that rechallenge with checkpoint inhibitors following initial injury can be safe [76,78,79]. This suggests that factors extrinsic to the treatment and patient genetics, such as the immune status of the patient or environmental factors, may also play a role in the development of hepatotoxicity in response to checkpoint inhibitors.

While it is reasonable that checkpoint inhibitors may cause IDILI by reducing inhibition of the immune response, the TNF-α inhibitors are indicated in many inflammatory autoimmune conditions like rheumatoid arthritis, in which they are expected to be immunosuppressive. It is therefore less intuitive that this class of biologics might also carry the risk of causing IDILI, but it is clear that they do. The presentation of liver injury mediated by anti-TNF-α agents often appears to be autoimmune-like itself, with the presence of autoantibodies and a predominance in females [80]. Less commonly, the pattern of liver injury can also be hepatocellular without features of autoimmunity or cholestatic injury [81]. Several possible explanations have been proposed for the mechanisms underlying anti-TNF-α associated liver injury, including suppression of cytotoxic T cell-mediated suppression of autoreactive B cells or suppression of regulatory T cell expansion, which could increase autoimmunity [82,83]. However, cross-reactivity appears to be low between agents in this class [84], which suggests that TNF-α inhibition itself may not be the mechanism (or the only mechanism) of injury, but perhaps the structures or specific effects of the individual molecules themselves. It has been speculated that infliximab carries a higher risk of causing liver injury due to the fact that it is a chimeric antibody [85]. Clearly, the mechanisms underlying anti-TNF-α therapy-related liver injury are complicated and likely multifaceted.

Rituximab is an anti-CD20 antibody that depletes B cells and is indicated for use in multiple malignancies and autoimmune conditions. While mild-moderate ALT elevations are fairly common during rituximab treatment (10–15%) [86], clinically apparent hepatotoxicity due to rituximab is quite rare [87]. The exception is hepatotoxicity due to latent viral reactivation, usually hepatitis B [88]. Although rituximab is also a chimeric mouse/human antibody like infliximab, the two antibodies clearly have different propensities to cause liver injury.

Daclizumab is an antibody against CD25, the alpha subunit of the IL-2 receptor. It was initially approved to treat and prevent acute cellular rejection following solid organ transplantation and was more recently approved to treat relapsing-remitting multiple sclerosis. In both cases, it is intended to be immunosuppressive by inhibiting lymphocyte activation. Daclizumab carried a black box warning for hepatotoxicity and was ultimately withdrawn from the market due to several cases of serious inflammatory brain disorders [89]. While daclizumab is expected to inhibit T cell activation, this also includes inhibition of regulatory T cells. Therefore, the effects of daclizumab could also involve immune activation in addition to suppression of cellular immunity.

Just as there are different patterns of liver injury in response to different small molecule drugs, there are different patterns of liver injury in response to different biologics. The mechanisms that underlie biologic-related IDILI are still largely unknown and may result from the interaction of multiple factors including their effect on the immune system and their structures.

## 3. Mechanistic Hypotheses

From the clinical presentations of IDILI described above, there are clearly common characteristics to the IDILI caused by different drugs. Primary among these common characteristics are their idiosyncratic nature, and the usual delay between starting the drug and the onset of liver injury. Such characteristics are typical of an immune reaction. Yet, there are variations on that theme with different drugs and even with the same drug in different patients. There is clear evidence of an immune mechanism for the IDILI caused by many drugs, and the commonality of characteristics suggest that most IDILI is immune mediated. There is no example of IDILI that is clearly not immune mediated; however, there are usually exceptions to any “rule”; therefore, it would not be surprising if there are cases of IDILI that are not immune mediated. Other mechanisms have been proposed for IDILI, but such hypotheses, if true, are more likely to be complementary to an immune mechanism rather than excluding an immune mechanism as discussed below.

### 3.1. Immune Mechanisms

There are multiple lines of evidence that indicate most IDILI is immune mediated. This evidence includes HLA associations [11], associations with other immune-related genes, clinical characteristics, and histology as detailed above in Section 2. Although not all IDILI has been linked to specific HLA haplotypes, this is often due to the availability of sufficient cases required to perform the genetic studies. In other cases, the lack of an HLA association may have to do with the nature of the reactive metabolite involved as discussed below. Given the evidence that most IDILI is immune mediated, the next question is how does a drug induce an immune response which, in some patients, leads to IDILI? As discussed below, in many cases, it appears to be a chemically reactive species, either the drug as in the case of flucloxacillin, or more commonly a reactive metabolite as in the case of halothane. Such reactive species modify proteins and other macromolecules, and this results in “foreign” molecules or neoantigens. Neoantigens can be taken up by antigen presenting cells, processed to peptides, and presented in the context of major histocompatibility complex-II (MHC-II, the human form of MHC is HLA) to CD4+ helper T cells (Th) that express a T cell receptor that can “recognize” this peptide. This is referred to as signal 1 of T cell activation. However, this signal will not activate the T cell unless the antigen presenting cell has been activated and expresses costimulatory molecules such as CD80, CD86, and CD40. This co-stimulation is referred to as signal 2. The presence of cytokines also influences T cell activation and is referred to as signal 3. Signal 3 is required for proliferation of the T cells that recognize the drug-modified peptides, and it also determines what type of helper T cell is produced. The major types of helper T cells are Th1 cells that promote a cell-mediated adaptive immune response and Th2 cells that promote an antibody-mediated adaptive immune response. It is, of course, more complex, and there are other types of CD4+ T cells including, but not limited to, Th17 cells that are generally proinflammatory, and Tregs that promote immune tolerance.

Signal 1 and signal 3 are relatively straight forward. An important question is what generates signal 2. The danger hypothesis posits that unless a foreign substance causes some type of injury or cell stress, the immune system will ignore it [90]. Cell injury can cause the release of damage-associated molecular pattern molecules (DAMPs). DAMPs act through pattern recognition receptors (PRRs) such as Toll-like receptors to activate antigen presenting cells. There are many molecules that can act as DAMPs including HMGB1, heat shock proteins, ATP, etc., [91]. A major mode of transfer of DAMPs from where they are formed to immune cells appears to be exosomes [92].

Although CD4+ helper cells are required for an immune response, the major effector cells are CD8+ cytotoxic T cells and B cells, the latter of which become plasma cells that produce antibodies following activation. Most immune responses are a combination of cell- and antibody-mediated immune response, and anti-drug or autoantibodies are sometimes observed in cases of IDILI; however, the histology of most IDILI suggests that it is the cell-mediated immune response that is responsible for most of the liver damage [93]. In addition, as mentioned above, neoantigens formed in hepatocytes by reactive metabolites are presented in the context of MHC-I (HLA class-I in humans) because, while all nucleated cells express MHC-I, MHC-II (HLA class II in humans) expression is largely restricted to antigen presenting cells. MHC-I binds to CD8, which is consistent with the histology of IDILI that is characterized by numerous CD8+ T cells [93]. In addition, many of the HLA associations that predict the risk of IDILI associated with a specific drug are HLA class I alleles [11], which again suggests that a Th1 and CD8+ cell-mediated adaptive immune response causes most of the liver injury in IDILI. It is interesting that patients who develop peripheral eosinophilia or liver eosinophilia appear to have a milder IDILI course [36]. Eosinophils are dependent on IL-5, which is a Th2 cytokine, and the milder course may be due to a greater Th2 rather than Th1 type of immune response.

If drug-modified proteins are presented in the context of MHC-I, which binds to CD8, how do T helper cells, which express CD4 and bind to MHC-II, become activated? In general, antigen presenting MHC-II+ cells do not contain significant concentrations of cytochromes P450, which is the major enzyme involved in the production of reactive metabolites. Antigen presenting cells could take up apoptotic hepatocytes and present fragments of them in the context of MHC-II; however, the turnover of hepatocytes is generally quite low. The drug could cause enough cell damage to increase the rate of hepatocyte turnover, but most drugs that cause IDILI do not appear to cause much overt toxicity. Under inflammatory conditions, hepatocytes can express MHC-II, but most IDILI appears to occur in the absence of a major inflammatory stimulus. The most likely explanation is that the hepatocytes release the drug-modified proteins in exosomes, which are taken up by antigen presenting cells and presented in the context of MHC-II. Exosomes appear to represent a major mechanism for communication between cells, and drug-modified proteins have been detected in exosomes [94]. This represents the most likely mechanism for activation of T helper cells. Exosomes can also contain DAMPs, and this could represent a mechanism for activation of antigen presenting cells [92].

If most IDILI is mediated by the adaptive immune system, it provides an easy answer to the question of what makes IDILI idiosyncratic. If the immune response involves drug-modified proteins, then there must be drug-modified peptides that are recognized by MHC-I, MHC-II, and T cell receptors on both helper T cells and effector T cells. Similar to the association of idiosyncratic drug reactions with specific HLA haplotypes, there is also evidence for an association with specific T cell receptors [95]; however, this is more difficult to study. The number of different MHC-I molecules expressed by an individual is 6, and that for MHC-II is 8. This limited number of MHC molecules is presumably why the IDILI associated with several drugs is associated with specific HLA haplotypes. In contrast, the number of possible T cell receptors is almost limitless because they are the product of random gene recombination. However, the number of T cells is not unlimited, and the number of T cells stays relatively constant throughout life. Therefore, if new memory T cells are produced in response to a new pathogen, other memory T cells have to die to make room for them, and the T cell repertoire of an individual changes over time in response to what they are exposed to. Structurally different antigenic molecules can bind to the same T cell receptor by binding in different orientations and sites of interaction. This is referred to a heterologous immunity, and this greatly increases the number of antigenic molecules that can be recognized [96]. However, it can also result in unanticipated cross reactivity to structurally different molecules. Thus, if a drug-modified peptide is recognized by a memory T cell that is the result of a response to a pathogen, it can increase the immune response to the corresponding drug [97]. There is evidence that drug reaction with eosinophilia and systemic symptoms (DRESS) involves cross reactivity between memory cells specific for human herpes virus and the drugs that cause DRESS, and DRESS is frequently associated with reactivation of the virus [98]. This appears to be because the immune response in DRESS includes an increase in T regulatory cells (Treg) to decrease tissue damage, and this allows the virus to proliferate.

The immune response is a delicate balance between an active immune response and immune tolerance. Antigen recognition is relative; the stronger the affinity of the drug-modified peptide for the T cell receptor and HLA, the stronger the immune response is likely to be, and the less co-stimulation required to overcome immune tolerance. In addition, memory T cells that have been primed by a pathogen are likely to result in a stronger immune response, as appears to be the case of DRESS as described above. It is likely that almost everyone has MHC-I, MHC-II, and T cell receptors that have some affinity for one or more of the many drug-modified peptides that can be formed by the reactive metabolite of a drug, or the drug itself if no chemically reactive species is involved. Even when there is a strong HLA association with IDILI caused by a specific drug, most patients will not develop IDILI when treated with that drug. In addition, with many drugs there will be some patients who do not have that HLA haplotype associated with an increased risk and still develop IDILI [16]. Each immune response likely involves multiple T cells with varying affinities for the drug or drug-modified peptide.

The immune response is extremely complex with many checks and balances to prevent the immune response from causing too much damage to the host. In particular, the dominant immune response in the liver is immune tolerance because it is exposed to many inflammatory molecules from the intestine, and a strong immune response to all of these molecules would cause liver damage [99]. There are multiple mechanisms that limit the immune response including Tregs and myeloid-derived suppressor cells, cytokines such as IL-10, and other molecules including immune checkpoints such as PD-1 and CTLA-4. Even when drugs produce liver injury, it more commonly resolves despite continued treatment. This is referred to as adaptation, but it presumably represents immune tolerance. Exosomes also appear to be important in transferring molecules involved in immune tolerance [100]. In the PD-1^ȡ/−^ impaired immune tolerance model described above, we have shifted the balance away from immune tolerance, which resulted in significant liver injury caused by several drugs that cause IDILI in humans. However, without high affinity binding of the drug-modified peptide to MHC and T cell receptors, the immune response was limited, and it did not result in liver failure. In fact, with most drugs the injury resolved despite continued treatment, again presumably because of immune tolerance. Immune checkpoint inhibitors also increase the risk of IDILI caused by co-administered drugs in humans [101], but again it is a delicate balance, and the risk does not increase to 100%. This may also be why inflammatory diseases such as Crohn’s disease do not appear to markedly increase the risk of IDILI, i.e., such inflammatory conditions presumably also increase factors that promote immune tolerance to limit the damage done by the disease. There are probably many factors that cause immune dysregulation that can influence the risk that a patient will develop IDILI [16].

Although most of the liver damage appears to be mediated by the adaptive immune system, activation of the adaptive immune system requires an innate immune response to activate antigen presenting cells and produce inflammatory cytokines. While the adaptive immune response is idiosyncratic because it requires specific MHC and T cell receptors, the innate immune response is activated by reactive metabolites and DAMPs and is unlikely to be idiosyncratic. For example, clozapine can cause idiosyncratic agranulocytosis and IDILI. Although most patients do not have an idiosyncratic reaction to clozapine, most patients do have an innate immune response, often with a transient increase in serum IL-6 levels and a paradoxical neutrophilia [102]. We find a similar response in rats treated with clozapine [103]. We also found that the supernatant from hepatocyte spheroids incubated with nevirapine activated inflammasomes in antigen presenting cells [104]. This is presumably because the hepatocytes released DAMPs into the media. The ability of a drug to produce a clinically silent innate immune response in patients, and even in animals, may provide a way to predict which drug candidates will cause a significant risk of IDILI when many patients are treated. The proposed steps in the immune response to drugs that can cause IDILI are summarized in Figure 1.

### 3.2. Reactive Metabolites

There is a large amount of circumstantial evidence to suggest that most IDILI is caused by reactive metabolites [105]. The fact that most drug metabolism occurs in the liver provides a reason for the liver to be a common site of idiosyncratic drug reactions. Most reactive metabolites are formed by cytochromes P450, although other enzymes can also form reactive metabolites. Reactive metabolites provide an obvious source of signal 1, and they can also cause cell damage or stress leading to the production of signal 2. The association between covalent binding and the risk of IDILI led some drug companies to screen drug candidates for their degree of covalent binding [106]. It was found that correcting for the daily dose of the drug increased the correlation between the amount of covalent binding and IDILI risk; however, the correlation was far from perfect [107]. By definition, covalent binding leads to neoantigens, which can generate signal 1; however, it is likely that different reactive metabolites vary in their ability to cause the release of DAMPs that can generate signal 2. The ability of a drug to cause mild injury and the release of DAMPS may be the most important variable that determines whether a reactive metabolite is likely to cause IDILI. For example, it appears that reactive metabolites such as acyl glucuronides are less likely to cause IDILI than other reactive metabolites; however, this is controversial [108]. In addition, other characteristics of the covalent binding may also be important. For example, the major reactive metabolite of isoniazid binds to lysine amino groups and appears to covalently bind to most proteins [31]. This will lead to a very large number of drug-modified peptides that can be recognized by the immune system. Moreover, acetylhydrazine and hydrazine, two metabolites of isoniazid, are also metabolized to reactive metabolites, which also would likely react with proteins to form different drug-modified proteins. This may be why no strong HLA associations have been found for the risk of isoniazid IDILI, and an increase in ALT occurs in about 20% of patients treated with isoniazid. Other reactive metabolites are “softer” and react selectively with thiols. If a reactive metabolite is less reactive, non-covalent interactions may also determine which proteins are modified. For example, the reactive metabolites of clozapine and vesnarinone are “soft” electrophiles, which react selectively with thiol nucleophiles. They also are both bioactivated by myeloperoxidase and have similar half-lives, at least in solution. Therefore, it would be expected that they would modify similar proteins. However, we found that the spectrum of modified proteins was quite different, presumably because non-covalent interactions prior to the irreversible covalent binding significantly affected to which proteins they bind [109]. If the number of proteins that a reactive metabolite modifies is limited, it would limit the number of drug-modified peptides that could bind to MHC proteins, and this may make it more likely that there will be a strong HLA association with the risk of IDILI for that drug.

### 3.3. Non-Covalent Interactions: The p-i Hypothesis and Modified Binding of Endogenous Peptides

Although there is a large amount of circumstantial evidence that reactive metabolites are involved in the mechanism of IDILI, there is little direct evidence. It is hard to imagine that halothane IDILI does not involve a reactive metabolite. However, it is always dangerous to extrapolate very far from direct evidence, and association does not prove causation. In addition, there are a few drugs such as ximelagatran that cause IDILI but do not appear to form a reactive metabolite. Therefore, it is likely that even if reactive metabolites are responsible for most IDILI, there are likely to be exceptions. Even if ximelagatran does not form a reactive metabolite, the mechanism of ximelagatran IDILI appears to be immune mediated because it is associated with HLA -DRB1*07 and HLA-DQA1*02. These are MHC-II molecules. As mentioned, the dominant immune response in the liver is immune tolerance. If ximelagatran does not form a reactive metabolite, why does it cause liver injury and not some other type of idiosyncratic drug reaction such as a skin rash? In fact, ximelagatran did appear to cause a skin rash in some workers involved in the production of the drug [4]. In addition, there were hints that ximelagatran is concentrated in the liver, presumably by a transporter. In fact, many drugs are concentrated in the liver, and this is likely another factor that makes the liver a frequent target of idiosyncratic drug reactions.

If ximelagatran does not form a reactive metabolite that can generate signal 1, how does it cause IDILI? There are two related hypotheses that have been proposed by which a non-covalent interaction can induce an immune response, i.e., the p-i hypothesis and the modified binding of endogenous peptide hypotheses. The p-i hypothesis proposes that non-covalent interactions with the MHC/T cell receptor complex can initiate an immune response [110]. This is based on the observation that in some cases, peripheral mononuclear cells from patients with a history of an idiosyncratic reaction to a drug can be activated by the parent drug in the absence of any metabolism. However, this assumes that the cells respond to the same molecules as what induced the immune response in the first place. Although this sounds plausible, it is not always true. For example, nevirapine causes an immune mediated skin rash, specifically in female Brown Norway rats, that is very similar to the rash that it causes in humans [55]. This is one case in which it could be clearly demonstrated that a specific reactive metabolite is responsible for an idiosyncratic drug reaction because the reactive metabolite responsible is a reactive benzylic sulfate formed in the skin, and a topical sulfotransferase inhibitor prevented the covalent binding and rash where it was applied [54]. Even though we know that a reactive metabolite is responsible, peripheral mononuclear cells from sensitized animals respond to the parent drug with the production of INF-γ [111]. In a similar way, cells from patients with a history of overt isoniazid IDILI respond to the parent drug even though it is very likely that reactive metabolites are responsible for the liver injury [34]. In contrast, cells from patients with a history of transient increases in ALT only responded to drug-modified proteins. This is presumably because a strong immune response leads to proliferation of T cells with many different T cell receptors, and in some cases, there are T cells that recognize the parent drug even though it was a drug-modified protein that initiated the immune response. However, these findings do not falsify the p-i hypothesis, it just means that one must be cautious.

A related mechanism was discovered in the study of abacavir-induced hypersensitivity reactions. It was found that there is a very strong association between the risk this idiosyncratic reaction and HLA-B*5701. It was also found that abacavir binds very tightly to this MHC-I molecule and changes the endogenous peptides to which MHC-I binds [112,113]. Since the immune system has never “seen” these peptides before, they are recognized as foreign. That can account for signal 1, but it is less clear what produces signal 2, although abacavir does form a reactive metabolite [114]. The mechanism of abacavir-induced hypersensitivity appears to be uncommon. Ximelagatran has a structure similar to a peptide and appears to have a high affinity for the two associated HLA molecules [115]. This may explain the mechanism of ximelagatran IDILI, but there is no direct evidence to confirm this mechanism. An HLA-DR7 and HLA-DQ2 transgenic mouse model was generated and treated with ximelagatran, but no liver injury was observed [116]. A plausible explanation is that even though the presence of ximelagatran may change the peptides that would bind to these MHC-II molecules, the mice did not have the T cell receptors required to recognize them.

Pyrazinamide and allopurinol cause IDILI with clear HLA associations, but they also do not appear to form chemically reactive metabolites. Their structural resemblance to purine and pyrimidine bases, respectively, may be involved, but that is highly speculative.

### 3.4. BSEP Inhibition

A genetic deficiency in bile salt export protein (BSEP, ABCB11) is associated with severe cholestatic liver injury resulting in liver failure in infancy [117]. The biochemical characteristics include a markedly elevated alkaline phosphatase and serum bile salts, but a normal γ-glutamyl transpeptidase. Other variants are much more benign even though there can be an almost complete absence of detectable BSEP [118]. It is stated that BSEP inhibition is one mechanism by which a drug can cause IDILI [119]; however, it has also been stated that in vitro measures of BSEP inhibition are not useful predictors of IDILI [120]. It was reported that a common polymorphism in BSEP (1331T > C → V444A) was observed more frequently in patients with cholestatic IDILI; however, most patients who developed cholestatic IDILI did not express this variant [121]. In addition, this was not observed in a Japanese population [122]. There are several other transporters and molecules involved in the control of bile salt production and transport as well as compensatory mechanisms that complicate the picture [123]. Another complication is that a metabolite, such as the sulfate conjugate of troglitazone, may have a greater effect on BSEP than the parent drug [124]. Unfortunately, virtually all of the data to indicate that BSEP inhibition is a significant factor in the mechanism of IDILI comes from in vitro studies at high concentrations of drug, and there is little clinical evidence to indicate that inhibition of BSEP is significant in patients. One obvious experiment would be to determine if drugs that inhibit BSEP in vitro produce an increase in bile salts in patients who take the drug. One paper by Fattinger et al., found that the mean level of serum bile salts increased when patients treated with bosentan developed liver injury, but it did not occur in all patients, and it was not clear that the increase occurred before the onset of liver injury [125]. In the same paper, co-treatment with glyburide, which also inhibits BSEP, appeared to increase the incidence of bosentan-induced liver injury. Many of the drugs that inhibit BSEP are associated with a hepatocellular type of liver injury rather than cholestatic as would be expected if the injury were due solely to BSEP inhibition. It is more likely that BSEP inhibition is one of many factors that can cause cell stress resulting in the release of DAMPs and contribute to immune mediated liver injury in susceptible patients. However, we simply do not have sufficient clinical evidence to determine the significance of BSEP inhibition in the mechanism of IDILI.

### 3.5. Mitochondrial Injury

It is clear that valproate IDILI involves mitochondria as described above. Another example is fialuridine, a drug developed to treat hepatitis B. Unfortunately, it inhibited the synthesis of mitochondrial DNA in humans, which resulted in serious toxicity involving the liver and other organs. The liver injury was characterized by lactic acidosis and microvesicular steatosis, which is consistent with mitochondrial injury [126]. However, this liver injury was not idiosyncratic and involved other organs as well. Other reverse transcriptase inhibitors can also cause mitochondrial injury leading to effects on various organs including the liver [127]. Mitochondria are also a key target in the mechanism of acetaminophen-induced liver injury [128]. Linezolid can also cause adverse reactions, including lactic acidosis, which apparently involves the inhibition of mitochondrial protein synthesis [129]. It has been proposed that inhibition of the mitochondrial electron transport chain is one mechanism by which drugs can cause IDILI, and in vitro assays to test for this effect have been used to screen drug candidates for the risk that they will cause IDILI [2]. The classic drugs that inhibit the mitochondrial electron transport chain are the biguanides, phenformin and metformin, and their major serious toxicity is lactic acidosis. However, they rarely, if ever, cause liver injury. It was reported that the combination of rotenone, the classic agent used to inhibit the first step in the mitochondrial electron transport chain, and isoniazid, a drug associated with a significant risk of IDILI, caused hepatocyte cell death in vitro at concentrations that alone were not toxic [130]. Based on this observation, it was proposed that drugs that inhibit the mitochondrial electron transport chain could act synergistically in patients to cause IDILI. When we tested this combination in vivo in mice it was lethal. When we decreased the dose of rotenone so that it was not lethal, it did not lead to liver injury [47]. When we tested the combination in our impaired immune tolerance model, the addition of rotenone did not increase the immune mediated liver injury caused by isoniazid. From these results, we conclude that inhibition of the mitochondrial electron transport chain is not a significant mechanism of IDILI. Probably the more important observation is that metformin not only does not cause IDILI, it does not appear to increase the risk of IDILI caused by co-administered drugs. That does not mean that other types of mitochondrial injury do not play a role in the mechanism of IDILI. Most IDILI does not have features characteristic of mitochondrial injury such as lactic acidosis and microvesicular steatosis. That suggests that when mitochondria are involved in the mechanism of IDILI, mitochondrial injury is not the sole mechanism. On the other hand, mitochondria can be critical sources of reactive oxygen production and source of DAMPs that promote an immune response [131]. It is likely that mitochondria play a role in the mechanism of some IDILI other than valproate, but we have virtually no clear data as to how often or exactly what role they play.

### 3.6. Unfolded Protein Response

Covalent binding of a drug to proteins has the potential to change their conformation and lead to an unfolded protein response. This can cause cell stress, which in turn, could promote activation of an immune response [132]. It has been proposed that the unfolded protein response plays a role in the mechanism of IDILI. We performed a preliminary in vivo experiment with a drug that undergoes extensive covalent binding in the liver and found no evidence of the unfolded protein response. This should not be considered significant evidence against the hypothesis; however, it did make us question the hypothesis. Even though drug covalent binding is easy to detect with a good anti-drug antibody, the absolute amount of covalent binding is very small, and it may not be significant relative to other causes of unfolded protein such as viral infections [132]. Therefore, this also remains an unsubstantiated hypothesis.

### 3.7. Reactive Oxygen Species/Oxidative Stress

Reactive oxygen species can cause liver injury [133], and this has been proposed as a mechanism for IDILI [134]. Most of the reactive metabolites that we study are formed by two electron oxidations to form electrophiles that can covalently bind to proteins. However, many nitrogen-containing drugs can also undergo one electron oxidations to form free radicals that are unlikely to covalently bind to proteins, but they can initiate free radical chain reactions and formation of reactive oxygen species. Probably the best examples are drugs that contain a primary aromatic amine functional group. Examples include sulfonamide antibiotics, dapsone, aminoglutethimide, and procainamide. Although these drugs are associated with a variety of idiosyncratic drug reactions, with the exception of sulfamethoxazole, IDILI is uncommon. Other drugs such as chlorpromazine also have the potential to produce free radicals in addition to electrophilic reactive metabolites [135]. Oxidative stress has the potential to cause the release of DAMPs, although there are many protective mechanisms, especially in the liver, that minimize the damage caused by such species. Oxidative stress can be produced by neutrophils and mitochondria [136]. A heterozygous superoxide dismutase 2 model was produced in which the animals developed delayed onset liver injury when treated with troglitazone [137]; however, others were not able to reproduce this model [138]. Bile acids also appear to cause oxidative stress [139]. As with many other plausible hypotheses, it is untested and there are insufficient data to link reactive oxygen species to the mechanism of IDILI.

## 4. Clinical Challenges

The ultimate goal is to minimize the impact of IDILI on patients. This could be accomplished by either prevention or effective treatment of IDILI. A better understanding the mechanisms of IDILI is likely to facilitate achieving these goals.

### 4.1. Diagnosis

The first challenge is the diagnosis of IDILI. IDILI can mimic other types of liver injury, and if cases are misdiagnosed, it can lead to false conclusions about which drugs can cause IDILI and what characterizes IDILI caused by specific drugs. Unfortunately, there is no test that can be used to differentiate IDILI from other forms of liver injury [140]. The two most common methods that have been used to diagnose IDILI are a panel of 3 experts to adjudicate cases and the Roussel Uclaf Causality Assessment Method (RUCAM) [141]. RUCAM has the advantages of being objective and not requiring experts; however, it has the disadvantage that it cannot easily incorporate characteristics of the IDILI caused by different drugs. Since there is no test that can provide a definitive diagnosis, it is not possible to compare the accuracy of the two methods. The mechanisms of liver injury produced by different causes are likely to be similar; therefore, it may never be possible to develop a simple blood test to differentiate IDILI from other causes of liver injury. Theoretically, a test specific to the drug such as a lymphocyte transformation test could differentiate IDILI from other causes of liver injury. However, the lymphocyte transformation test has never been standardized or validated, and it appears to be associated with a high number of false negatives. If the immune response is directed against drug-modified proteins rather than the parent drug it is not surprising that there would be false negatives. Therefore, even with a better mechanistic understanding of IDILI, it may be impossible to develop a simple and reliable test to diagnosis IDILI; it would be better to focus on prevention and treatment.

### 4.2. Prevention

There are two ways to prevent IDILI: develop drugs that are not associated with a significant incidence of IDILI (design safer drugs), or since most patients that take a drug that can cause IDILI do not develop significant liver injury, predict which patients will actually develop IDILI (design safer patients).

As discussed above, it is likely that most IDILI is immune mediated or at least has a significant immune component. However, due to the patient-specific risk factors that contribute to the idiosyncrasy of the severe adaptive immune system-mediated reactions, they are not usually detected during preclinical development. As mentioned, although there are many methods that have been used to screen drug candidates to predict risk, none have proven to be accurate. It is likely that drugs have to cause some type of mild cell damage in order to release DAMPs and induce an immune response. Drugs are dirty, i.e., they have many biological effects other than the desired therapeutic effect, e.g., BSEP inhibition, and it is likely that it would be almost impossible to screen drug candidates for all the effects that may cause the release of DAMPs or otherwise influence the risk that a drug candidate will be associated with a relatively high IDILI risk. In addition, although it is a plausible hypothesis, it has not been clearly demonstrated that BSEP inhibition is linked to IDILI risk. A better strategy would be to study downstream events that may be common to drugs that cause IDILI. Assays to measure the release of DAMPs might be one way to screen drug candidates. However, there are many different types of DAMPs, and it is not yet clear if there are different patterns of DAMP release that may be associated with an increased IDILI risk. Downstream of the release of DAMPs is an innate immune response, which is required for the idiosyncratic adaptive immune response that actually causes liver injury. As mentioned earlier, the innate immune response to drugs that can cause IDILI is unlikely to be idiosyncratic; therefore, it could be studied in patients who will not develop IDILI, and in most cases, even in animals. It is clear that some drugs that cause IDILI do produce a clinically silent innate immune response in most patients and animals as discussed above, but this has received very little attention. Further research may reveal that there are common innate immune pathways that accurately predict IDIL risk. There may be just a few such pathways that could even be screened with in vitro assays such as the activation of inflammasomes by the supernatant from an incubation of a drug with macrophages [104]. However, it may be quite complex with many different mechanisms by which a drug can produce an innate immune response, and it may require in vivo models that measure several innate immune parameters to detect the range of patterns of innate immune response activation. To validate the precision and reliability of such an approach, it would first require extensive investigation of how drugs that are known to cause IDILI affect the innate immune response in both humans as well as relevant animal models. Since different drugs may cause different innate immune responses, it would require the study of multiple drugs that cause IDILI as well as the study of drugs that are not associated with a significant risk of IDILI. The study of the immune response to drugs is likely to make the process of drug development safer, but it is not a trivial task.

With respect to designing safer patients, one way to predict which patients may be at an increased risk for developing IDILI is genetic screening for implicated HLA haplotypes. While this has been demonstrated to be effective in some cases, such as screening for the HLA-B*57:01 haplotype to prevent abacavir hypersensitivity [142], this is not practical in most cases, and many drugs associated with the risk of IDILI do not have known HLA associations. Few other genetic markers or environmental factors have been found to substantially increase risk, and none could practically be used clinically. It is possible that the intensity of an early innate immune response would predict which patients would later have a strong adaptive immune response that leads to IDILI. We saw this with an animal model of penicillamine-induced autoimmunity, which included liver injury. Even though the animals were inbred, only half of the treated animals developed the syndrome after 3 weeks of treatment, and although all of the animals had a transient increase in IL-6 one day after starting penicillamine, the serum IL-6 was much higher in the animals that later developed autoimmunity and liver injury [143]. However, it seems unlikely that such a method would work in most cases or be practical.

Another method that may markedly decrease risk is slow dose titration. Dose titration has been proposed as a general method to improve drug safety [144], but this may be of special importance in the case of idiosyncratic drug reactions such as IDILI. Again, using the penicillamine model, a low dose of penicillamine for 2 weeks completely prevented the autoimmune syndrome and liver injury; one week of low dose treatment was not sufficient [145]. This was immune tolerance because it could be transferred to naïve animals with spleen cells from tolerized animals. Dose titration decreases the risk of idiosyncratic reactions to lamotrigine [146] and nevirapine [147], but it is not completely protective. It is possible that a more gradual dose escalation would be more effective in preventing idiosyncratic reactions such as IDILI. In recent clinical trials of cenobamate there were 3 cases of drug reaction with eosinophilia and systemic symptoms (DRESS) with one death. It was recommended that the dose be started at 12.5 mg/day based on the drugs given at a dose of 10 mg/day or less are unlikely to cause serious idiosyncratic drug reactions [10]. The dose was then doubled every 2 weeks until a therapeutic dose of about 200 mg/day was achieved [148]. This appeared to overt a significant risk of DRESS, and cenobamate was approved in late 2019 with a mandate for this strict dosing regimen [149]. It is possible that this practice could be extended to prevent IDILI; however, it is necessary to test this hypothesis with additional high-risk drugs. Ultimately, achieving any of these goals first requires a more comprehensive understanding of the basic mechanisms underlying IDILI.

### 4.3. Treatment

The first step in the treatment of IDILI is discontinuation of the drug involved. In most cases the patient will recover, and no specific treatment is required. In fact, if the IDILI is very mild, the patient may “adapt”, and it may be possible to treat through the liver injury if the drug is considered essential. Presumably, this adaptation represents immune tolerance. However, serious IDILI often progresses after the drug is stopped. This is made obvious by the fact that halothane IDILI is not clinically evident until about a week after halothane exposure, and there are examples where the onset of IDILI occurs more than 3 weeks after drug treatment has been stopped [4]. In principle, this provides a window in which treatment may prevent the development of liver failure. There is some evidence that N-acetylcysteine may be helpful even in DILI cases that are not caused by acetaminophen [150]. Ursodeoxycholic acid is often used to treat cholestatic IDILI and carnitine is used to treat valproate IDILI [151]. However, such treatments are not very effective for preventing liver failure. If IDILI is mediated by the adaptive immune system, immunosuppression might be effective. Corticosteroids are often used, but there is no evidence that they decrease mortality or liver transplantation. One study of the effect of steroids found no benefit; however, the study involved liver failure patients with a variety of etiologies [152]. Steroids have not proven to be effective in other types of immune mediated idiosyncratic reactions such as toxic epidermal necrolysis [153]. The one exception is autoimmune IDILI, which responds well to corticosteroids [154]. Corticosteroids are also recommended for IDILI caused by immune checkpoint inhibitors [155]. If the most serious cases of IDILI are mediated by CD8+ T cells, then agents that target these cells could be effective. Cyclosporin does appear to decrease the mortality associated with toxic epidermal necrolysis, which is also mediated by CD8+ T cells [156]. Another idiosyncratic drug reaction that is mediated by T cells is aplastic anemia. It is known that antithymocyte globulin and cyclosporin are effective for the treatment of aplastic anemia whether it is idiopathic or drug-induced [157]. Mycophenylate and cyclosporin have been suggested for treatment of IDILI caused by immune checkpoint inhibitors, but there is little data to indicate whether they are effective or not [155]. A very interesting therapeutic possibility is the use of Tregs, which can be expanded ex vivo [158]. The development of an effective treatment for serious IDILI would certainly be welcomed; however, it would be difficult to perform the clinical trial required to demonstrate that it is effective.

## 5. Conclusions

There are multiple lines of evidence that most IDILI is mediated by the adaptive immune system, especially CD8+ T cells. Presumably, the major factors that make IDILI idiosyncratic is the requirement for HLA class I and class II and T cell receptors that have a high affinity for the drug, or more likely, one of the many drug-modified peptides formed by a reactive metabolite of the drug. It is likely that everyone has HLA class I and class II and T cell receptors that will recognize one of the many drug-modified peptides that can be produced; however, the affinity can vary widely and have a large effect on the balance between whether a patient develops IDILI or immune tolerance. In addition, memory T cells that have been primed by a pathogen that happen to cross react with the drug or drug-modified peptides are more likely to lead to serious IDILI. Other factors that can affect this balance include genes that affect the immune response, immune checkpoint inhibitors, or other factors that cause immune dysregulation.

The adaptive immune response requires an innate immune response to generate signals 2 and 3. The innate immune response is likely caused by some type of mild cell damage or stress that leads to the release of DAMPs. This cell damage may also be caused by a reactive metabolite, but it could also be caused by a variety of biochemical effects of the drug such as BSEP inhibition. The innate immune response is likely to be clinically silent but unlikely to be idiosyncratic. There are often differences between humans and animals, but many aspects of drug metabolism and immune response are similar; therefore, the innate immune response to a drug may often be similar in animals, as we see in the case of clozapine. This would allow careful studies of the innate immune response to drugs, which could then be extended to humans to make certain that observations represent mechanisms in humans. The immune response is extremely complex with many checks and balances and involves many different cytokines, chemokines, and other molecules, as well as many types of immune cells that change their phenotype over time. Our current understanding of the immune response to drugs is very superficial, and the study of the early steps in the immune response to drugs could have a profound effect on our understanding of the mechanisms of IDILI. This mechanistic understanding, in turn, is likely to markedly facilitate drug development and improve drug safety.

## Figures and Tables

**Figure 1 ijms-22-02954-f001:**
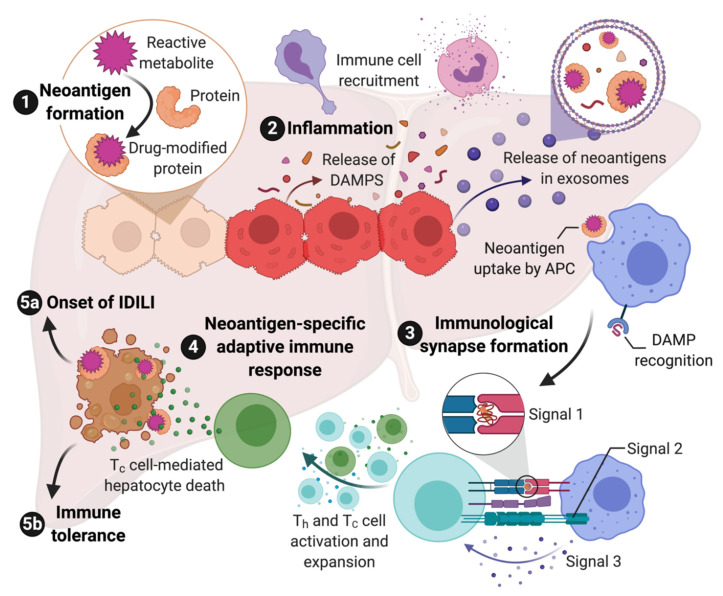
Proposed general mechanism of idiosyncratic drug-induced liver injury (IDILI). The following are the proposed sequence of steps in the mechanism of IDILI: In order to produce an immune response, the drug must interact with MHC-II to produce signal 1 (the human form of MHC is HLA). (**1**) In most cases this involves the formation of a reactive metabolite by hepatocytes that covalently binds to proteins. These modified proteins can act as neoantigens that are presented by antigen presenting cells (APCs) that express MHC-II. These neoantigens are released by hepatocytes, probably packaged in exosomes. However, there are examples in which the drug does not appear to form a reactive metabolite and may interact non-covalently with MHC-II or the MHC-T cell receptor complex. In order to produce an immune response, the drug must also activate antigen presenting cells. (**2**) This involves the release of damage-associated molecular pattern molecules (DAMPs) such as heat shock proteins and HMGB1. The production of DAMPs can also be caused by a reactive metabolite that causes cellular dysfunction, although in some cases the release of DAMPs may involve some other mechanism such as inhibition of the bile salt export pump by the parent drug. The DAMPs are released from hepatocytes, most likely in exosomes, and activate antigen presenting cells through pattern recognition receptors. The DAMPs also lead to the recruitment of other innate immune cells. (**3**) The neoantigens are taken up by antigen present cells to produce signal 1, and activation of these cells by DAMPs leads to expression of costimulatory molecules such as CD80 and CD40, which provide signal 2 to CD4+ helper T cells. (**4**) The helper T cells are activated by the combination of signal 1 and signal 2 provided by the antigen presenting cell. The helper T cells produce cytokines that facilitate and shape the immune response. Th1 helper T cells promote a cell-mediated adaptive immune response, and Th2 helper T cells promote an antibody-mediated adaptive immune response. Most adaptive immune responses are a combination of both cell and antibody immune responses. (**5a**) However, because the dominant presentation of intracellular antigens such as the neoantigens produced by reactive metabolites is through MHC-I, which binds to CD8 on cytotoxic T cells, the dominant adaptive immune response in IDILI is usually a cell-mediated immune response. The first steps in this mechanism likely occur in most patients; however, unless the drug, or more likely drug-modified peptides, are recognized by MHC-II, MHC-I, and T cell receptors, there will be no adaptive immune response and no liver injury. The activated antigen presenting cells can also activate CD4+ Treg that dampen the immune response as well as release of cytokines such as IL-10 that also dampen the immune response. (**5b**) Therefore, unless the binding of the drug, or more likely a drug-modified peptide is very strong, the adaptive immune response will end in immune tolerance, which prevents or limits liver injury.

## Data Availability

Not applicable.

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
