# Peer review of "Idiosyncratic Drug-Induced Liver Injury: Mechanistic and Clinical Challenges"

_ijms, 2021, doi:10.3390/ijms22062954_

Round 1
Reviewer 1 Report
As announced in the title, the manuscript is a review of the current knowledge on idiosyncratic DILI mechanisms. However, the clinical challenges are just touched upon without a discussion on the clinical aspects recognized as the main issues in practice. The hypotheses and the description of the mechanisms are nonetheless those developed by the team from the 90's without any surprise. Which is reassuring.
Here are my comments:
1- Among the clinical challenges, limited to paragraph 2 (page2), there is nothing about causality assessment which is the main practical issue in IDILI. There is no one word on RUCAM the widely used causality assessment method in the world. This part of the manuscript should be expanded in order to be consistent with the title.
2- Inaccuracies or mistakes are unexpected with regard to the scientific quality of the team.
2.1- The example of telithromycin (Ref 5) for a very short time interval (1 or 2 days) between the first intake of the offending drug and the jaundice is inappropriate. Indeed in the 3 cases described in the article the jaundice occurred 1 or 2 days after the first intake and this is pathophysiologically incompatible. Even in paracetamol overdose where the liver injury is quite fast and massive, the jaundice appears after 2 or 3 days and this without an obvious immune system involvement. The 3 patients might have had an underlying liver disease at the time of telithromycin administration. The exclusion of alternative causes was minimal and notoriously insufficient. Moreover the authors used a causality assessment method in 10 questions (WHO or Naranjo?) clearly not adapted to DILI. RUCAM should have been used. It is therefore inadequate to take this example of IDILI with short time to onset and this should be changed in the manuscript.
The other example could be (quote) "some fluoroquinolones" but the authors did not provide a reference. This lack of reference should be corrected.
Anyway, in IDILI there is no good example where the time to onset in very short. This would go against the immune system activation which needs several days to be activated except in case of drug rechallenge.
More generally the authors should stressed the point that using clinical cases of IDILI have to be validated by a strong method such as RUCAM updated in 2016 otherwise confusion will remain in research on IDILI.
2.2- The reference (3) for the definition of the types of liver injury is wrong. The authors refer to Chalasani et al (2015) while the definition was one of the Conclusions of an international consensus meeting published in J Hepatol1990; 11: 272-6 and well known by experts in DILI. This mistake should be corrected.
2.3- There is a conceptual confusion in the classification of liver injury: the ratio R defines 3 biochemical types of liver injury (hepatocellular, cholestatic and mixed) rightly named but not the auto immune "type" which is a mechanism that could be biochemically expressed by any of the 3 types. The manuscript should be corrected accordingly.
2.4- The lymphocyte transformation test is now generally considered as too sensitive and almost not specific to determine the cause of the DILI. Of note, the years of references (22,32) : 1997 and 1978 respectively, nothing more recent or with a critical opinion on this test. It is not (and should not be) used as an evidence of the immune system involvement but only that the drug or its metabolite is recognised by lymphocytes not necessarily as the cause of the liver injury. The value of this test should be correctly discussed or this part should be deleted.
3 - Strikingly, among the molecular mechanisms the role of CYPs is not really explained in any of the examples while one of the authors stressed this point elsewhere (Teschke R, Uetrecht J. Mechanism of idiosyncratic drug induced liver injury (DILI): Unresolved basic issues. In special issue: Unresolved basic issues in hepatology, guest editors Ralf Weiskirchen & Wolfgang Stremmel. Ann Transl Med 2021. View this article at: http://dx.doi.org/10.21037/atm-2020-ubih-05 ). Does that mean that its role is marginal in the production of reactive metabolites/changes in proteins at the very origin of the immune system activation? For some drugs auto antibodies are directed against CYPs involved in their own metabolism. Is this important fact insignificant in the mechanism? In the abstract CYPs are absent. Focusing only on the immune system would reduce the role of other important steps in our understanding of the mechanisms and would put apart a large part of the research in this area to help us prevent IDILI. These aspects should be pointed out in the manuscript to respect a fair balance between the mechanisms.
Otherwise, the manuscript nicely describes various hypotheses and authors opinion on IDILI mechanisms that are all based on the immune system activation by changes in native proteins as a consequence of drug metabolism. This is likely true based on experiments showing the production of reactive metabolites or ROS involving CYPs or mitochondria damage depending on drug metabolism pathways. The main issue in nonclinical and clinical research remains the prediction of IDILI.
It must be admitted that despite years of research in this area, based on animal experiments, in vitro studies and clinical trials, the clinicians are still unable to prevent IDILI.
Reviewer 2 Report
The main idea of the manuscript is interesting, but in some parts the text is no more than a simple and extended description of other authors and does not contribute directly to the publication.
Please modify the next:
Abstract: clarify the main objectives of the review (and follow this goal thought the entire manuscript)
Point 2. Inform the criteria of drug selection and discuss more in detail the relation of the drugs elected with IDILI.
2.2 and following: define briefly the mechanism of action of the drugs and discuss a probably relation with liver drug metabolism.
I Would like to know the incidence of the elected drugs in IDILI, instead to read phrases like “it is well known…” or “one of the first drugs to be associated…”
2.6 the first paragraph is not related to valproate, please modify it.
What happens with chlorpromazine, and acetaminophen? They are drugs with enough information about liver injury.
Line 325 there is an extra space
It would be interesting to discuss more profoundly the role of infliximab.
For line 415 to 436; 475 to 487; 490 to 495; 496 to 504 and so some more down in the text: resume the ideas and discuss it in relation to the main objective, or at least In relation with the knowing of immunology response in liver injury (in IDILI)
The authors have too much textual cites and some of them are no clearly focus in the main idea of the manuscript.
Please attach a table that relates the drugs with the HLAs and other imunologic information
Round 2
Reviewer 2 Report
The authors must make corrections to the writing accordingly to what was prevously requested.